behaviour, evolution, theoretical biology

cooperation, altruism, assortment, cultural evolution, evolutionary models, Prisoner's Dilemma

**Author for correspondence:**
Yoav Ram
e-mail: yoav@yoavram.com

# Non-vertical cultural transmission, assortment and the evolution of cooperation

Dor Cohen[1], Ohad Lewin-Epstein[2], Marcus W. Feldman[5] and Yoav Ram[1,3,4]

[1]School of Computer Science, Interdisciplinary Center Herzliya, Herzliya, Israel
[2]School of Plant Sciences and Food Security, Faculty of Life Sciences, [3]School of Zoology, Faculty of Life Sciences, and [4]Sagol School of Neuroscience, Tel Aviv University, Tel Aviv, Israel
[5]Department of Biology, Stanford University, Stanford, CA, USA

OL-E, 0000-0002-8636-9006; MWF, 0000-0002-0664-3803; YR, 0000-0002-9653-4458

Cultural evolution of cooperation under vertical and non-vertical cultural transmission is studied, and conditions are found for fixation and coexistence of cooperation and defection. The evolution of cooperation is facilitated by its horizontal transmission and by an association between social interactions and horizontal transmission. The effect of oblique transmission depends on the horizontal transmission bias. Stable polymorphism of cooperation and defection can occur, and when it does, reduced association between social interactions and horizontal transmission evolves, which leads to a decreased frequency of cooperation and lower population mean fitness. The deterministic conditions are compared to outcomes of stochastic simulations of structured populations. Parallels are drawn with Hamilton's rule incorporating relatedness and assortment.

## 1. Introduction

Cooperative behaviour can reduce an individual's fitness and increase the fitness of its conspecifics or competitors [1]. Nevertheless, cooperative behaviour appears to occur in many animals [2], including humans, primates [3], rats [4], birds [5,6] and lizards [7]. Evolution of cooperative behaviour has been an important focus of research in evolutionary theory since at least the 1930s [8]. Since the work of Hamilton [9] and Axelrod & Hamilton [1], theories for the evolution of cooperative and altruistic behaviours have been intertwined often under the rubric of *kin selection*. Kin selection theory posits that natural selection is more likely to favour cooperation between more closely related individuals. The importance of *relatedness* to the evolution of cooperation and altruism was demonstrated by Hamilton [9], who showed that an allele which determines cooperative behaviour will increase in frequency if the reproductive cost to the actor that cooperates, $c$, is less than the benefit to the recipient, $b$, times the relatedness, $r$, between the recipient and the actor. This condition is known as *Hamilton's rule*:

$$c < b \times r, \tag{1.1}$$

where the relatedness coefficient $r$ measures the probability that an allele sampled from the cooperator is identical by descent to one at the same locus in the recipient.

Eshel & Cavalli-Sforza [10] studied a related model for the evolution of cooperative behaviour. Their model included *assortative meeting*, or non-random encounters, where a fraction $m$ of individuals in the population each interact specifically with an individual of the same phenotype, and a fraction $1 - m$ interacts with a randomly chosen individual. Such assortative meeting may be owing, for example, to population structure or active partner choice.

In their model, cooperative behaviour can evolve if ([10], eqn (3.2))

$$c < b \times m, \quad (1.2)$$

where $b$ and $c$ are the benefit and cost of cooperation.[1]

The role of assortment in the evolution of altruism was emphasized by Fletcher & Doebeli [11, p. 15]. They found that in a *public-goods* game, altruism will evolve if cooperative individuals experience more cooperation, on average, than defecting individuals, and 'thus, the evolution of altruism requires (positive) assortment between focal *cooperative* players and cooperative acts in their interaction environment.' With some change in parameters, this condition is summarized by ([11], eqn (2.3))

$$c < b \times (p_C - p_D), \quad (1.3)$$

where $p_C$ is the probability that a cooperator receives help, and $p_D$ is the probability that a defector receives help.[2] Bijma & Aanen [12] obtained a result related to inequality (1.3) for other games.

Cooperation can also evolve when interactions are determined by population structure. For example, Ohtsuki *et al.* [13] studied populations on graphs with average degree $k$, that is, the average individual has $k$ potential interaction partners. Assuming that selection is weak and that the population size is much larger than $k$ (i.e. sparse structure), they found that cooperative behaviour can evolve if

$$c < b \times \frac{1}{k}. \quad (1.4)$$

They thus interpret $1/k$ as *social relatedness* or *social viscosity* [13].

Cooperative behaviour can be subject to *cultural transmission*, which allows an individual to acquire attitudes or behavioural traits from other individuals in its social group through imitation, learning, or other modes of communication [14,15]. Feldman *et al.* [16] introduced the first model for the evolution of altruism by cultural transmission with kin selection and demonstrated that if the fidelity of cultural transmission of altruism is $\varphi$, then the condition for evolution of altruism in the case of sibling-to-sibling altruism is ([16], eqn (16))

$$c < b \times \varphi - \frac{1-\varphi}{\varphi}. \quad (1.5)$$

In inequality (1.5), $\varphi$ replaces relatedness ($r$ in inequality (1.1)) or assortment ($m$ in inequality (1.2)), but the effective benefit $b \times \varphi$ is reduced by $(1-\varphi)/\varphi$. This shows that under cultural transmission, the condition for the evolutionary success of altruism entails a modification of Hamilton's rule (inequality (1.1)).

Cultural transmission may be modelled as vertical, horizontal or oblique: vertical transmission occurs between parents and offspring, horizontal transmission occurs between individuals from the same generation, and oblique transmission occurs to offspring from the generation to which their parents belong (i.e. from non-parental adults). Evolution under either of these non-verticle transmission models can be more rapid than under pure vertical transmission [14,17,18]. Both Woodcock [19] and Lewin-Epstein *et al.* [20] demonstrated that non-vertical transmission can help explain the evolution of cooperative behaviour, the former using simulations with cultural transmission, the latter using a model where cooperation is mediated by host-associated microbes. Indeed, models in which microbes

affect their host's behaviour [20–22] are mathematically similar to models of cultural transmission, which also emphasize the role of non-vertical transmission [14].

Here, we study models for the cultural evolution of cooperation that include both vertical and non-vertical transmission. In our models, behavioural changes are mediated by cultural transmission that can occur specifically during social interactions. For instance, there may be an association between the choice of partner for social interaction and the choice of partner for cultural transmission, or when an individual interacts with an individual of a different phenotype, exposure to the latter may lead the former to convert its phenotype. Our results demonstrate that cultural transmission, when associated with social interactions, can favour the evolution of cooperation even when genetic transmission cannot, partly because it facilitates the generation of assortment [11], and partly because it diminishes the effect of selection (owing to non-vertical transmission from non-reproducing individuals [18]).

## 2. Models

Consider a very large well-mixed population whose members can be one of two phenotypes: $\phi = A$ for cooperators or $\phi = B$ for defectors. An offspring inherits its phenotype from its parent via vertical cultural transmission with probability $v$ or from a random individual in the parental population via oblique transmission with probability $(1 - v)$ (figure 1a). Following Ram *et al.* [18], given that the parent's phenotype is $\phi$ and assuming uni-parental inheritance [23], the conditional probability that the phenotype $\phi'$ of the offspring is $A$ is

$$P(\phi' = A \mid \phi) = \begin{cases} v + (1-v)p, & \text{if } \phi = A \\ (1-v)\,p, & \text{if } \phi = B \end{cases}, \quad (2.1)$$

where $p = P(\phi = A)$ is the frequency of $A$ among all adults in the parental generation.

Not all adults become parents, and we denote the frequency of phenotype $A$ among parents by $\dot{p}$. Therefore, the frequency $\hat{p}$ of phenotype $A$ among juveniles (after selection and vertical and oblique transmission) is

$$\hat{p} = \dot{p}[v + (1-v)p] + (1-\dot{p})[(1-v)p] = v\dot{p} + (1-v)p. \quad (2.2)$$

Individuals are assumed to interact according to a *Prisoner's Dilemma*. Specifically, individuals interact in pairs; a cooperator suffers a fitness cost $0 < c < 1$, and its partner gains a fitness benefit $b$, where we assume $c < b$ (i.e. donation game). Figure 1c shows the pay-off matrix: the fitness of an individual with phenotype $\phi_1$ when interacting with a partner of phenotype $\phi_2$.

Social interactions occur randomly: two juvenile individuals with phenotype $A$ interact with probability $\hat{p}^2$, two juveniles with phenotype $B$ interact with probability $(1 - \hat{p})^2$, and two juveniles with different phenotypes interact with probability $2\hat{p}(1 - \hat{p})$. Horizontal cultural transmission occurs between pairs of individuals from the same generation. It occurs between socially interacting partners with probability $\alpha$, or between a random pair with probability $1 - \alpha$ (figure 1b). However, horizontal transmission is not always successful, as one partner may reject the other's phenotype. The probability of successful horizontal transmission of phenotypes $A$ and $B$ are $T_A$ and $T_B$, respectively (table 1 and figure 1d). Thus, the frequency $p'$ of phenotype

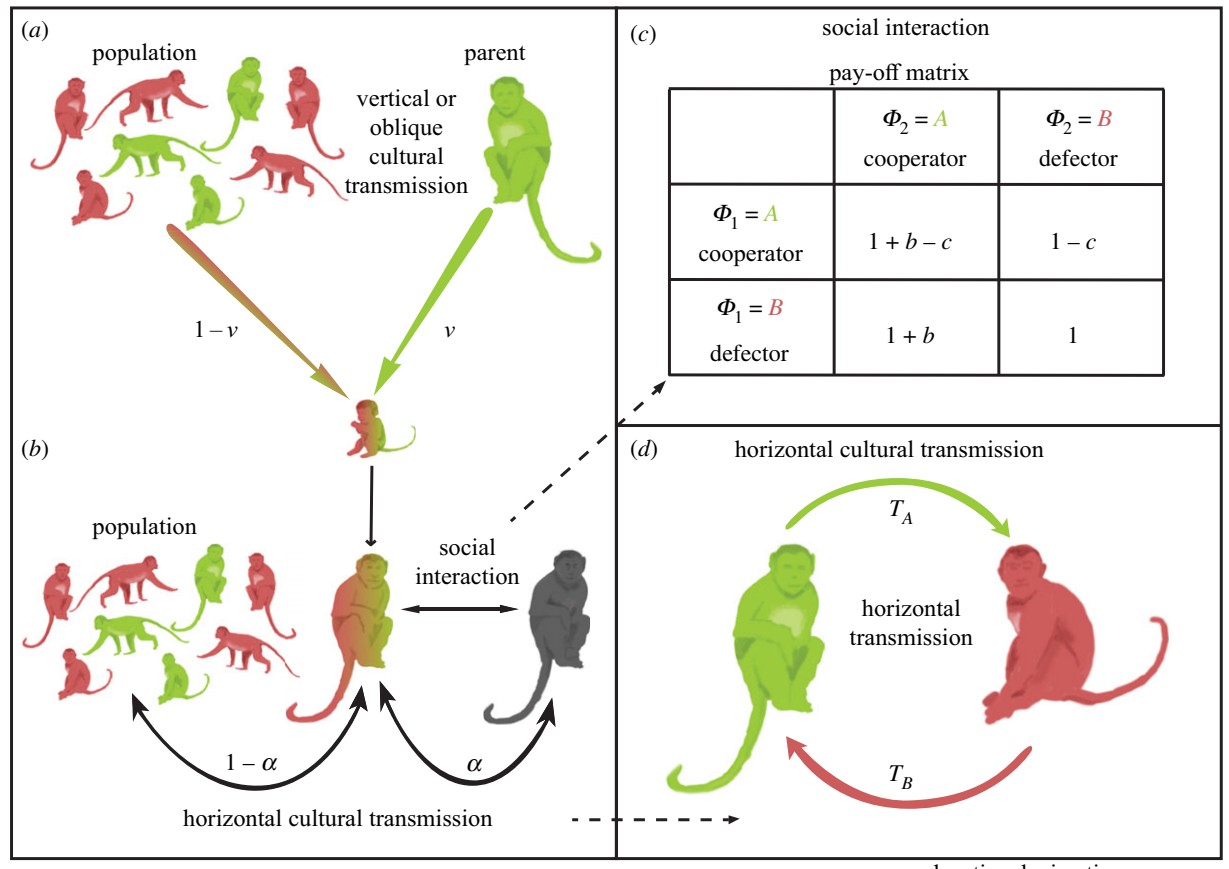

**Figure 1.** Model illustration. (*a*) First, offspring inherit their parent's phenotype via vertical cultural transmission with probability $v$, or the phenotype of a random non-parental adult via oblique cultural transmission with probability $1 - v$. (*b*) Second, adults socially interact in pairs in a Prisoner's Dilemma game. Horizontal cultural transmission occurs from a random individual in the population, with probability $1 - \alpha$, or from the social partner, with probability $\alpha$, where $\alpha$ is the interaction-transmission association parameter. (*c*) The Prisoner's Dilemma pay-off matrix shows the fitness of phenotype $\phi_1$ when interacting with phenotype $\phi_2$. (*d*) The probabilities of successful horizontal cultural transmission of phenotypes $A$ (cooperator) and $B$ (defector) are $T_A$ and $T_B$, respectively. (Online version in colour.)

**Table 1.** Interaction frequency, fitness and transmission probabilities.

| phenotype $\phi_1$ | phenotype $\phi_2$ | frequency | fitness of $\phi_1$ | $P(\phi_1 = A)$ via horizontal transmission: | |
| --- | --- | --- | --- | --- | --- |
| | | | | from partner, $\alpha$ | from population, $(1 - \alpha)$ |
| $A$ | $A$ | $\hat{p}^2$ | $1 + b - c$ | $1$ | $\hat{p} + (1 - \hat{p})(1 - T_B)$ |
| $A$ | $B$ | $\hat{p}(1 - \hat{p})$ | $1 - c$ | $1 - T_B$ | $\hat{p} + (1 - \hat{p})(1 - T_B)$ |
| $B$ | $A$ | $\hat{p}(1 - \hat{p})$ | $1 + b$ | $T_A$ | $\hat{p}T_A$ |
| $B$ | $B$ | $(1 - \hat{p})^2$ | $1$ | $0$ | $\hat{p}T_A$ |

*A* among adults in the next generation, after horizontal transmission, is

$$
\begin{aligned}
p' &= \hat{p}^2[\alpha + (1 - \alpha)(\hat{p} + (1 - \hat{p})(1 - T_B))] \\
&\quad + \hat{p}(1 - \hat{p})[\alpha(1 - T_B) + (1 - \alpha)(\hat{p} + (1 - \hat{p})(1 - T_B))] \\
&\quad + (1 - \hat{p})\hat{p}[\alpha T_A + (1 - \alpha)\hat{p}T_A] + (1 - \hat{p})^2[(1 - \alpha)\hat{p}T_A] \\
&= \hat{p}^2(T_B - T_A) + \hat{p}(1 + T_A - T_B).
\end{aligned}
\tag{2.3}
$$

For example, the first term in equation (2.3) describes the case where two juveniles with phenotype $A$ interact with probability $\hat{p}^2$. In this case, the focal individual will retain its phenotype if (i) its social interaction partner is also its horizontal transmission partner, with probability $\alpha$; or (ii) its horizontal transmission partner is another individual, with probability $(1 - \alpha)$, and (ii.a) that individual also has

phenotype $A$, with probability $\hat{p}$, or (ii.b) that individual has phenotype $B$, with probability $(1 - \hat{p})$, but horizontal transmission is unsuccessful, with probability $(1 - T_B)$. The frequency of $A$ among parents follows a similar dynamic but must also include the effect of natural selection. Therefore, each right-hand term from equation (2.3) is multiplied by the corresponding fitness value (table 1 and figure 1*c*), which depends on the phenotypes of the two interaction partners. Therefore, the frequency of phenotype $A$ among parents is

$$
\begin{aligned}
\bar{w}\dot{p}' &= \hat{p}^2(1 + b - c)[\alpha + (1 - \alpha)(\hat{p} + (1 - \hat{p})(1 - T_B))] \\
&\quad + \hat{p}(1 - \hat{p})(1 - c)[\alpha(1 - T_B) + (1 - \alpha)(\hat{p} + (1 - \hat{p})(1 - T_B))] \\
&\quad + (1 - \hat{p})\hat{p}(1 + b)[\alpha T_A + (1 - \alpha)\hat{p}T_A] \\
&\quad + (1 - \hat{p})^2[(1 - \alpha)\hat{p}T_A],
\end{aligned}
\tag{2.4}
$$

**Table 2.** Model variables and parameters.

| symbol | description | values |
|--------|-------------|--------|
| $A$ | cooperator phenotype | |
| $B$ | defector phenotype | |
| $p$ | frequency of phenotype $A$ among adults | [0, 1] |
| $\dot{p}$ | frequency of phenotype $A$ among parents | [0, 1] |
| $\hat{p}$ | frequency of phenotype $A$ among juveniles | [0, 1] |
| $v$ | vertical transmission rate | [0, 1] |
| $c$ | cost of cooperation | (0, 1) |
| $b$ | benefit of cooperation | $c < b$ |
| $\alpha$ | probability of interaction-transmission association | [0, 1] |
| $T_A, T_B$ | horizontal transmission rates of phenotype $A$ and $B$ | (0, 1) |

where fitness values are taken from figure 1c and table 1, and the population mean fitness is $\bar{w} = 1 + \hat{p}(b - c)$. Starting from equation (2.2) with $\hat{p}' = v\dot{p}' + (1 - v)p'$, we substitute $p'$ from equation (2.3) and $\dot{p}'$ from equation (2.4) and obtain

$$
\begin{aligned}
\hat{p}' =& \frac{v}{\bar{w}}\left[\hat{p}^2(1 + b - c)\left(1 - (1 - \hat{p})(1 - \alpha)T_B)\right)\right] \\
&+ \frac{v}{\bar{w}}\left[\hat{p}(1 - \hat{p})(1 - c)(\hat{p}(1 - \alpha)T_B + 1 - T_B)\right] \\
&+ \frac{v}{\bar{w}}\left[\hat{p}(1 - \hat{p})(1 + b)(\hat{p}(1 - \alpha) + \alpha)T_A\right] \\
&+ \frac{v}{\bar{w}}(1 - \hat{p})^2\hat{p}(1 - \alpha)T_A + (1 - v)\hat{p}^2(T_B - T_A) \\
&+ (1 - v)\hat{p}(1 + T_A - T_B).
\end{aligned}
\tag{2.5}
$$

Table 2 lists the model variables and parameters.

## 3. Results

We determine the equilibria of the model in equation (2.5) and analyse their local stability. We then analyse the evolution of a modifier of interaction-transmission association, $\alpha$. Finally, we compare derived conditions to outcomes of stochastic simulations with a structured population.

### (a) Evolution of cooperation

The fixed points (equilibria) of the recursion (equation (2.5)) are $\hat{p} = 0$, $\hat{p} = 1$, and (see the electronic supplementary material, eqn (B5))

$$
\hat{p}^* = \frac{\alpha bvT_A - cv(1 - T_B) + (T_A - T_B)}{[c(1 - v) - b(1 - \alpha v)](T_A - T_B)}.
\tag{3.1}
$$

Define the following cost thresholds, $\gamma_1$ and $\gamma_2$, and the vertical transmission threshold, $\hat{v}$,

$$
\begin{aligned}
\gamma_1 &= \frac{bv\alpha T_A + (T_A - T_B)}{v(1 - T_B)}, \\
\gamma_2 &= \frac{bv\alpha T_B + (1 + b)(T_A - T_B)}{v(1 - T_B) + (1 - v)(T_A - T_B)},
\end{aligned}
$$
and $\quad \hat{v} = \frac{T_B - T_A}{1 - T_A}$.
$\tag{3.2}$

Then we have the following result.

**Result 3.1.** With vertical, horizontal and oblique transmission, the cultural evolution of cooperation follows one of the following scenarios in terms of the cost thresholds $\gamma_1$ and $\gamma_2$ and the vertical transmission threshold $\hat{v}$ (equation (3.2)).

1. *Fixation of cooperation*: if *(i)* $T_A \geq T_B$ and $c < \gamma_1$; or if *(ii)* $T_A < T_B$ and $v > \hat{v}$ and $c < \gamma_2$.
2. *Fixation of defection*: if *(iii)* $T_A \geq T_B$ and $\gamma_2 < c$; or if *(iv)* $T_A < T_B$ and $\gamma_1 < c$.
3. *Stable polymorphism*: if *(v)* $T_A < T_B$ and $v < \hat{v}$ and $c < \gamma_1$; or if *(vi)* $T_A < T_B$ and $v > \hat{v}$ and $\gamma_2 < c < \gamma_1$.
4. *Unstable polymorphism*: if *(vii)* $T_A > T_B$ and $\gamma_1 < c < \gamma_2$.

Thus, cooperation can take over the population if it has either a horizontal transmission advantage, or if it has a horizontal transmission disadvantage but the vertical transmission rate is high enough. In either case, the cost of cooperation must be small enough. A stable polymorphism can exist between cooperation and defection only if defection has a horizontal transmission advantage. In this case, the existence of a stable polymorphism depends on the interplay between the benefit and cost of cooperation and the vertical transmission rate. These conditions are illustrated in figures 2a,b and 3a,b, and the analysis is in the electronic supplementary material, appendix B. Note that stable and unstable polymorphism are also called, respectively, *coexistence* and *bistable competition*.

Much of the literature on evolution of cooperation focuses on conditions for an initially rare cooperative phenotype to invade a population of defectors. The following remarks address this.

**Remark 3.2.** If the initial frequency of cooperation is very close to zero, then its frequency will increase if the cost of cooperation is low enough:

$$
c < \gamma_1 = \frac{bv\alpha T_A + (T_A - T_B)}{v(1 - T_B)}.
\tag{3.3}
$$

This merges the conditions for fixation of cooperation and for stable polymorphism, both of which entail instability of the state where defection is fixed, $\hat{p} = 0$.

Notably, increasing interaction-transmission association $\alpha$ increases the cost threshold ($\partial\gamma_1/\partial\alpha > 0$), making it easier for cooperation to increase in frequency when initially rare. Similarly, increasing the horizontal transmission of cooperation, $T_A$, increases the threshold ($\partial\gamma_1/\partial T_A > 0$), facilitating the evolution of cooperation (figure 3a,b). However, increasing the horizontal transmission of defection, $T_B$, can increase or decrease the cost threshold, but it increases the cost threshold when the threshold is already above one ($c < 1 < \gamma_1$): $\partial\gamma_1/\partial T_B$ is positive when $T_A > 1/(1 + \alpha bv)$, which gives $\gamma_1 > 1/v$. Therefore, increasing $T_B$ decreases the cost threshold and limits the evolution of cooperation, but only if $T_A < 1/(1 + \alpha bv)$.

Increasing the vertical transmission rate, $v$, can either increase or decrease the cost threshold, depending on the horizontal transmission bias, $T_A - T_B$, because $\text{sign}(\partial\gamma_1/\partial v) = -\text{sign}(T_A - T_B)$. When $T_A < T_B$, we have $\partial\gamma_1/\partial v > 0$, and as the vertical transmission rate increases, the cost threshold increases, making it easier for cooperation to increase when rare (figure 2b). By contrast, when $T_A > T_B$, we get $\partial\gamma_1/\partial v < 0$, and therefore as the vertical transmission rate increases, the

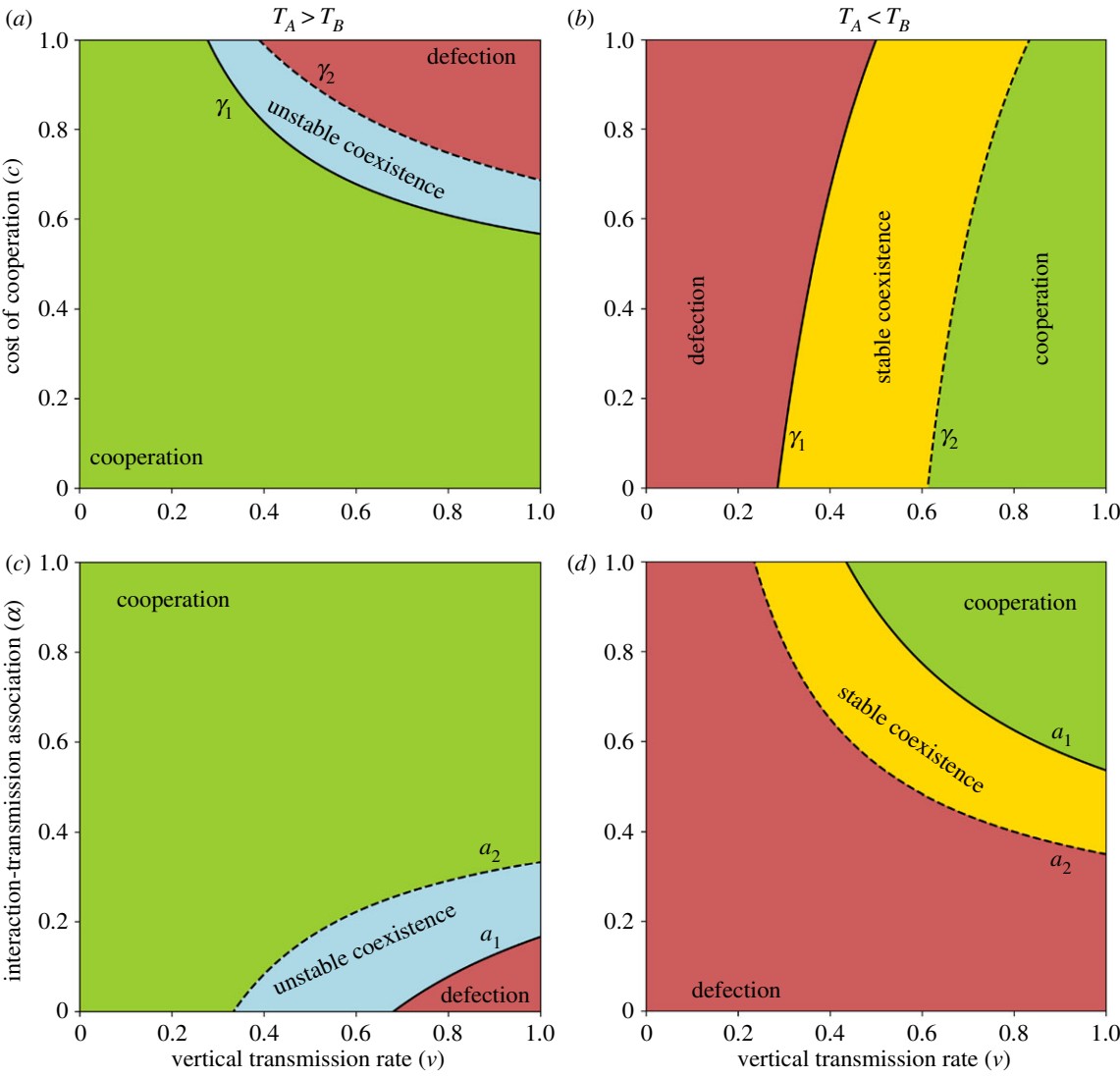

**Figure 2.** Evolution of cooperation under vertical, oblique and horizontal cultural transmission. The figure shows parameter ranges for global fixation of cooperation (green), global fixation of defection (red), fixation of either cooperation or defection depending on the initial conditions, i.e. unstable polymorphism (blue), and stable polymorphism of cooperation and defection (yellow). In all cases, the vertical transmission rate $v$ is on the $x$-axis. $(a,b)$ Cost of cooperation $c$ is on the $y$-axis and the cost thresholds $\gamma_1$ and $\gamma_2$ (equations (3.2)) are represented by the solid and dashed lines, respectively. $(c,d)$ Interaction-transmission association $\alpha$ is on the $y$-axis and the interaction-transmission association thresholds $a_1$ and $a_2$ (equation (3.7)) are represented by the solid and dashed lines, respectively. Horizontal transmission is biased in favour of cooperation, $T_A > T_B$, in $(a)$ and $(c)$, or defection, $T_A < T_B$, in $(b)$ and $(d)$. Here, $T_A = 0.5$, and $(a)$ $b = 1.2$, $T_B = 0.4$, $\alpha = 0.4$; $(b)$ $b = 2$, $T_B = 0.7$, $\alpha = 0.7$; $(c)$ $b = 1.2$, $T_B = 0.4$, $c = 0.5$; $(d)$ $b = 2$, $T_B = 0.7$, $c = 0.5$. (Online version in colour.)

cost threshold decreases, making it harder for cooperation to increase when rare (figure 2a).

Importantly, this condition cannot be formulated in the commonly used form of Hamilton's rule owing to the bias in horizontal transmission, represented by $T_A - T_B$. If $T_A = T_B = T$, then, from result 3.1 and inequality (3.3), cooperation will take over the population from any initial frequency if the cost is low enough:

$$c < b \times \frac{\alpha T}{1 - T},\tag{3.4}$$

regardless of the vertical transmission rate, $v$. This condition can be interpreted as a version of Hamilton's rule ($c < b \times r$, inequality (1.1)) or as a version of inequality (1.3), where $\alpha T/(1 - T)$ is a measure of *cultural relatedness* or *cultural assortment*, respectively, similar to the term *social relatedness* used by Ohtsuki *et al.* [13]. Note that the right-hand side of inequality (3.4) equals $\gamma_1$ when $T = T_A = T_B$.

From inequality (3.3), without interaction-transmission association ($\alpha = 0$), cooperation will increase when it is

rare if there is horizontal transmission bias for cooperation, $T_A > T_B$, and

$$c < \frac{T_A - T_B}{v(1 - T_B)}.\tag{3.5}$$

Figure 3a illustrates this condition (for $v = 1$), which is obtained by setting $\alpha = 0$ in inequality (3.3). In this case, the benefit of cooperation, $b$, does not affect the evolution of cooperation, and the outcome is determined only by cultural transmission. Furthermore, inequality (3.3) shows that with perfect interaction-transmission association ($\alpha = 1$), cooperation will increase when rare if

$$c < \frac{bvT_A + (T_A - T_B)}{v(1 - T_B)}.\tag{3.6}$$

In the absence of oblique transmission, $v = 1$, the only equilibria are the fixation states, $\dot{p} = 0$ and $\dot{p} = 1$, and cooperation will evolve from any initial frequency (i.e. $\dot{p}' > \dot{p}$) if inequality (3.6) applies (figure 3). This is similar to the case of microbe-induced

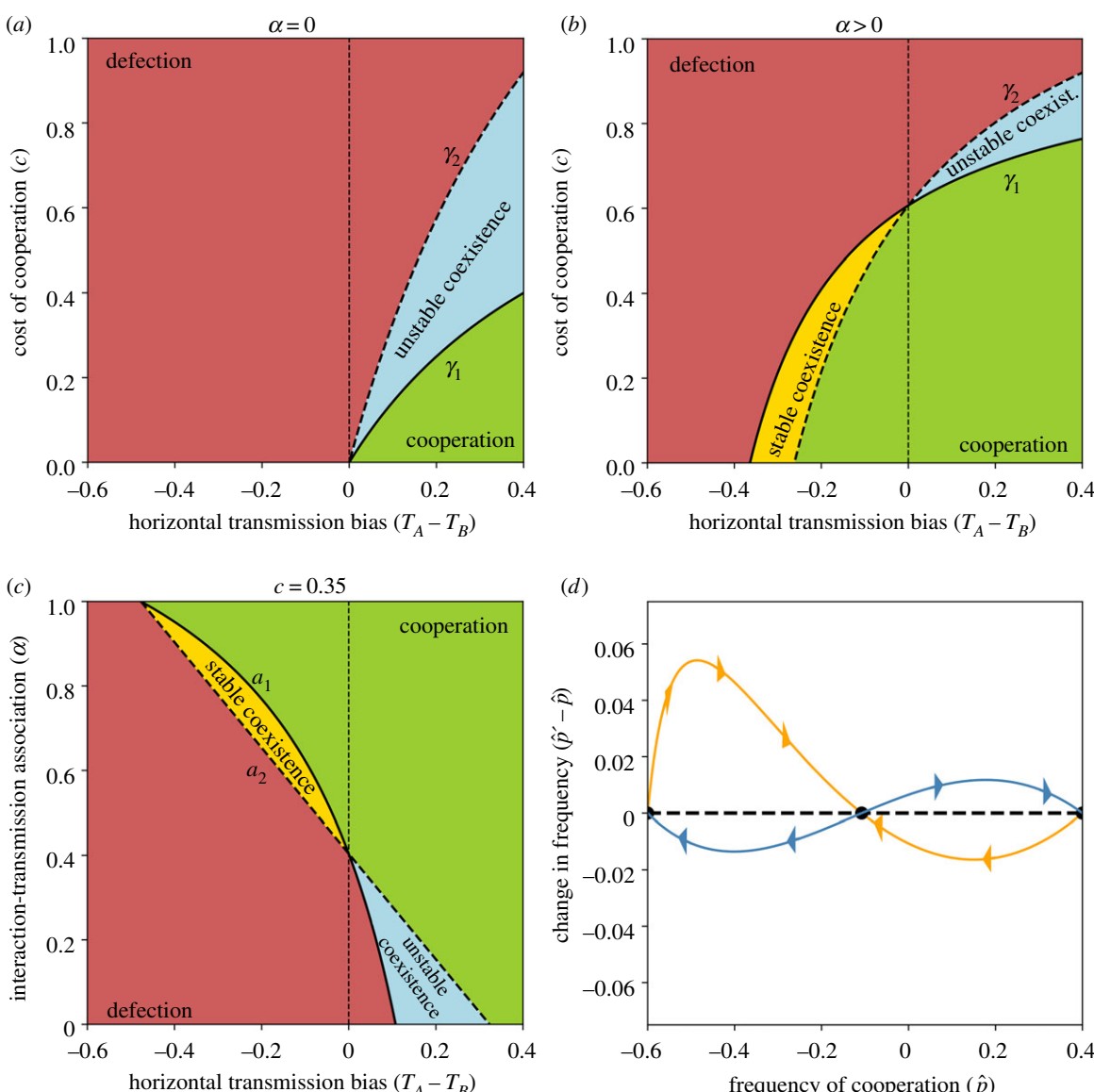

**Figure 3.** Evolution of cooperation under vertical and horizontal cultural transmission ($v = 1$). The figure shows parameter ranges for global fixation of cooperation (green), global fixation of defection (red), fixation of either cooperation or defection depending on the initial conditions, i.e. unstable polymorphism (blue), and stable polymorphism of cooperation and defection (yellow). (a–c) The horizontal transmission bias ($T_A - T_B$) is on the x-axis. In (a,b), the cost of cooperation $c$ is on the y-axis and the cost thresholds $\gamma_1$ and $\gamma_2$ (equation (3.2)) are the solid and dashed lines, respectively. In (c), interaction-transmission association $\alpha$ is on the y-axis and the interaction-transmission association thresholds $a_1$ and $a_2$ (equation (3.7)) are the solid and dashed lines, respectively. Here, $b = 1.3$, $T_A = 0.4$, $v = 1$, (a) $\alpha = 0$, (b) $\alpha = 0.7$, (c) $c = 0.35$. (d) Change in frequency of cooperation among juveniles ($\hat{p}' - \hat{p}$) as a function of the frequency ($\hat{p}$), see equation (2.5). The orange curve shows convergence to a stable polymorphism ($T_A = 0.4$, $T_B = 0.9$, $b = 12$, $c = 0.35$, $v = 1$ and $\alpha = 0.45$). The blue curve shows fixation of either cooperation or defection, depending on the initial frequency ($T_A = 0.5$, $T_B = 0.1$, $b = 1.3$, $c = 0.904$, $v = 1$ and $\alpha = 0.4$). Black circles show the three equilibria. (Online version in colour.)

cooperation studied by Lewin-Epstein *et al.* [20]; therefore, when $v = 1$, this remark is equivalent to their eqn (1.1).

It is interesting to examine the general effect of interaction-transmission association $\alpha$ on the evolution of cooperation. Define the interaction-transmission association thresholds, $a_1$ and $a_2$, as

$$a_1 = \frac{c \cdot v(1 - T_A) - (T_A - T_B)(1 + b - c)}{b \cdot v \cdot T_B}$$

and

$$a_2 = \frac{c \cdot v(1 - T_B) - (T_A - T_B)}{b \cdot v \cdot T_A}. \qquad (3.7)$$

**Remark 3.3.** Cooperation will increase when rare if interaction-transmission association is high enough, specifically if $a_2 < \alpha$.

Figure 2c,d illustrates this condition. With horizontal transmission bias for cooperation, $T_A > T_B$, cooperation can fix from any initial frequency if $a_2 < \alpha$ (green area right of dashed vertical line). With horizontal bias favouring defection, $T_A < T_B$, cooperation can fix from any frequency if $\alpha$ is large enough, $a_1 < \alpha$ (green area left of dashed vertical line), and can reach stable polymorphism if $\alpha$ is intermediate, $a_2 < \alpha < a_1$ (yellow area). Without horizontal bias, $T_A = T_B$, fixation of cooperation occurs if $\alpha$ is high enough, $\frac{c}{b} \times (1 - T)/T < \alpha$ (inequality (3.4); in this case $a_1 = a_2$).

Interestingly, because the sign of $\partial a_2 / \partial v$ is equal to the sign of $T_A - T_B$, the effect of the vertical transmission rate $v$ on $a_1$ and $a_2$ depends on the horizontal transmission bias. That is, if $T_A > T_B$, then evolution of cooperation is facilitated

by oblique transmission, whereas if $T_A < T_B$, then evolution of cooperation is facilitated by vertical transmission (figure 2*c*,*d*).

Next, we examine the roles of vertical and oblique transmission in the evolution of cooperation. Fixation of cooperation is possible only if the vertical transmission rate is high enough:

$$v > \hat{v} = \frac{T_B - T_A}{1 - T_A}. \tag{3.8}$$

This condition is necessary for fixation of cooperation, but it is not sufficient. If horizontal transmission is biased for cooperation, $T_A > T_B$, cooperation can fix with any vertical transmission rate (because $\hat{v} < 0$). By contrast, if horizontal transmission is biased for defection, $T_A < T_B$, cooperation can fix only if the vertical transmission rate is high enough: in this case oblique transmission can prevent fixation of cooperation (figure 2*b*,*d*).

With only vertical transmission ($v = 1$), from inequality (3.3), cooperation increases when rare if

$$c < \frac{b\alpha T_A + (T_A - T_B)}{1 - T_B}, \tag{3.9}$$

which can also be written as

$$\frac{c(1 - T_B) - (T_A - T_B)}{bT_A} < \alpha. \tag{3.10}$$

In the absence of vertical transmission ($v = 0$), from recursion (2.5), we see that the frequency of the cooperator phenotype among adults increases every generation, i.e. $p' > p$, if there is a horizontal transmission bias in favour of cooperation, namely $T_A > T_B$. That is, if $v = 0$, then selection plays no role in the evolution of cooperation (i.e. $b$ and $c$ do not affect $\hat{p}'$). The dynamics are determined solely by differential horizontal transmission of the two phenotypes. With no bias in horizontal transmission, $T_A = T_B$, phenotype frequencies do not change, $\hat{p}' = \hat{p}$.

Cooperation and defection can coexist at frequencies $\hat{p}^*$ and $1 - \hat{p}^*$ (equation (3.1)). When it is feasible, this equilibrium is stable or unstable under the conditions of result 3.1, parts 3 and 4, respectively. The yellow and blue areas in figures 2 and 3 show cases of stable and unstable polymorphism, respectively. When $\hat{p}^*$ is unstable, cooperation will fix if its initial frequency is $\hat{p} > \hat{p}^*$, and defection will fix if $\hat{p} < \hat{p}^*$. $\hat{p}^*$ is unstable when there is horizontal transmission bias for cooperation, $T_A > T_B$, and the cost is intermediate, $\gamma_1 < c < \gamma_2$. Figure 3*d* shows $\hat{p}' - \hat{p}$ as a function of $\hat{p}$.

## (b) Evolution of interaction-transmission association

We now focus on the evolution of interaction-transmission association under perfect vertical transmission, $v = 1$, assuming that the population is initially at a stable polymorphism of the two phenotypes, cooperation $A$ and defection $B$, where the frequency of $A$ among juveniles is $\hat{p}^*$ (equation (3.1)). Note that for a stable polymorphism, there must be horizontal bias for defection, $T_A < T_B$, and an intermediate cost of cooperation, $\gamma_2 < c < \gamma_1$ (equation (3.2)), see figure 3*b*. The equilibrium population mean fitness is $\bar{w}^* = 1 + \hat{p}^*(b - c)$, which is increasing in $\hat{p}^*$, and $\hat{p}^*$ is increasing in $\alpha$ (electronic supplementary material, appendix C). Therefore, $\bar{w}^*$ increases as $\alpha$ increases. But can this population-level advantage lead to the evolution of $\alpha$?

To answer this question, we add a *modifier locus* [24–27] that determines the value of $\alpha$ but has no direct effect on fitness. This locus has two alleles, $M$ and $m$, which induce interaction-transmission associations $\alpha_1$ and $\alpha_2$, respectively. Suppose that the population has evolved to a stable equilibrium $\hat{p}^*$ when only allele $M$ is present. We study the local stability of this equilibrium to invasion by the modifier allele $m$ (this is called *external stability* [26,28]) and obtain the following result.

**Result 3.4.** From a stable polymorphism between cooperation and defection, a modifier allele can successfully invade the population if it decreases the interaction-transmission association $\alpha$.

The analysis is in the electronic supplementary material, appendix D. This *reduction principle* [24,28] entails that successful invasions will reduce the frequency of cooperation, as well as the population mean fitness (electronic supplementary material, figure S1). Furthermore, if a modifier allele that decreases $\alpha$ appears and invades the population from time to time, then the value of $\alpha$ will continue to decrease, further reducing the frequency of cooperation and the population mean fitness. This evolution will proceed as long as there is a stable polymorphism, that is, as long as $a_2 < \alpha < a_1$ (remark 3.3; figure 3*c*). Thus, we can expect the value of $\alpha$ to approach $a_2$, the frequency of cooperation to fall to zero, and the population mean fitness to decrease to one (electronic supplementary material, figure S1). Note that $\alpha$ controls how often an individual learns from its interaction partner. However, from the *phenotype-centred view*, there is no incentive to do so: a cooperator interacting with a defector will not only pay the cost of cooperation but will also risk being 'converted' to defection (with probability $T_B$), whereas a defector interacting with a cooperator will forfeit (with probability $T_A$) the benefit it received.

## (c) Population structure

Interaction-transmission association may also emerge from population structure. Consider a population colonizing a two-dimensional grid of size 100-by-100, where each site is inhabited by one individual, similarly to the model of Lewin-Epstein & Hadany [21]. Each individual is characterized by its phenotype: either cooperator, $A$, or defector, $B$. Initially, each site in the grid is randomly colonized by either a cooperator or a defector, with equal probability. In each generation, half of the individuals are randomly chosen to 'initiate' interactions. Each initiator interacts (i) in a Prisoner's Dilemma game with a random neighbour (i.e. individual in a neighbouring site); and (ii) in horizontal cultural transmission with a random neighbour (with replacement, i.e. possibly the same neighbour). The expected number of each of these interactions per individual per generation is one, but the realized number of interactions can be zero, one, or even more than one, and in every interaction both individuals are affected, not just the initiator. The effective interaction-transmission association $\alpha$ in this model is the probability that the same neighbour is picked for both interactions, or $\alpha = 1/M$, where $M$ is the number of neighbours. On an infinite grid, $M = 8$ (i.e. Moore neighbourhood [29]), but on a finite grid $M$ can be lower in neighbourhoods close to the grid border. As before, $T_A$ and $T_B$ are the probabilities of successful horizontal transmission of phenotypes $A$ and $B$, respectively.

The order of the interactions across the grid at each generation is random. After all interactions take place, an

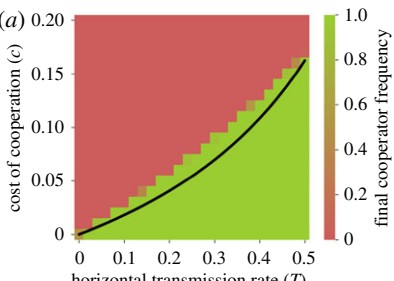
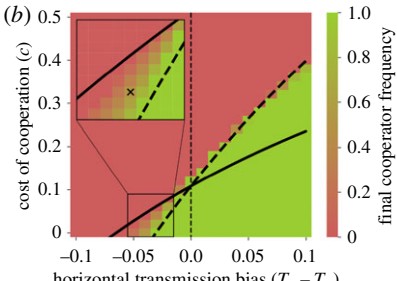
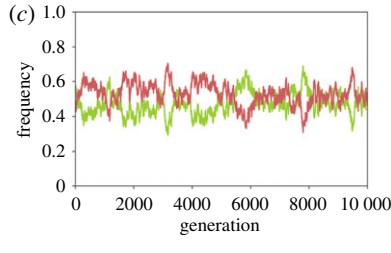

**Figure 4.** Evolution of cooperation in a structured population. (*a,b*) The expected frequency of cooperators in a structured population after 10 000 generations is shown (red for 0%, green for 100%) as a function of both the cost of cooperation, *c*, on the *y*-axis, and either the symmetric horizontal transmission rate, $T = T_A = T_B$, on the *x*-axis of panel (*a*), or the transmission bias, $T_A - T_B$, on the *x*-axis of panel (*b*). Black curves represent the cost thresholds for the evolution of cooperation in a well-mixed population with interaction-transmission association, where $\alpha = 1/8$ in inequality (3.4) for (*a*) and also in equations (3.2) for (*b*). The inset in (*b*) focuses on an area of the parameter range in which neither phenotype is fixed throughout the simulation, maintaining a stochastic locally stable polymorphism [30]. This stochastic polymorphism is illustrated in panel (*c*), which shows the frequency of cooperators (green) and defectors (red) over time for the parameter set marked by an *x* in (*b*). In all cases, the population evolves on a 100-by-100 grid. Cooperation and horizontal transmission are both local between neighbouring sites, and each site has eight neighbours. Selection operates globally (see the electronic supplementary material, figure S2 for results from a model with local selection). Simulations were stopped at generation 10 000 or if one of the phenotypes fixed. Fifty simulations were executed for each parameter set. Benefit of cooperation, $b = 1.3$; perfect vertical transmission $v = 1$. (*a*) Symmetric horizontal transmission, $T = T_A = T_B$; (*b*) horizontal transmission rate $T_A$ is fixed at 0.4, and $T_B$ varies, $0.3 < T_B < 0.5$; (*c*) horizontal transmission rates $T_A = 0.4 < T_B = 0.435$ and cost of cooperation $c = 0.02$. (Online version in colour.)

individual's fitness is determined by $w = 1 + (b \times n_b - c \times n_c)/n_i$ where $n_b$ is the number of interactions that individual had with cooperative neighbours, $n_c$ is the number of interactions in which that individual cooperated and $n_i$ is the total number of interactions in which that individual participated (note that the phenotype may change between consecutive interactions owing to horizontal transmission). Then, a new generation is produced, and the sites can be settled by offspring of any parent, not just the neighbouring parents. Selection is global, rather than local, in accordance with our deterministic model: the parent is randomly drawn with probability proportional to its fitness, divided by the sum of the fitness values of all potential parents. Offspring are assumed to have the same phenotype as their parents (i.e. $v = 1$).

The outcomes of stochastic simulations with such a structured population are shown in figure 4, which demonstrates that the highest cost of cooperation $c$ that permits the evolution of cooperation agrees with the conditions derived above for our model without population structure or stochasticity. An example of stochastic stable polymorphism is shown in figure 4*c*. Changing the simulation so that selection is local (i.e. sites can only be settled by offspring of neighbouring parents) had only a minor effect on the agreement with the derived conditions (electronic supplementary material, figure S2).

These comparisons show that the conditions derived for the deterministic unstructured model can be useful for predicting the dynamics in stochastic and structured models. Moreover, this structured population model demonstrates that our parameter for interaction-transmission association, $\alpha$, can represent local interactions between individuals.

## 4. Discussion

Under a combination of vertical, oblique and horizontal transmission with pay-offs in the form of a Prisoner's Dilemma game, cooperation or defection can either fix or coexist, depending on the relationship between the cost and benefit of cooperation, the horizontal transmission bias, and the association between social interaction and horizontal transmission (result 3.1; figures 2 and 3). Importantly, cooperation

can increase when initially rare (i.e. invade a population of defectors) if and only if (rewriting inequality (3.3)) $c \times v (1 - T_B) < b \times v \alpha T_A + (T_A - T_B)$, namely, the effective cost of cooperation (left-hand side) is smaller then the effective benefit plus the horizontal transmission bias (right-hand side). This condition cannot be formulated in the form of Hamilton's rule, $c < b \times r$, owing to the effect of biased horizontal transmission, represented by $(T_A - T_B)$. Remarkably, a polymorphism of cooperation and defection can be stable if horizontal transmission is biased in favour of defection ($T_A < T_B$) and both $c$ and $\alpha$ are intermediate (yellow areas in figures 2 and 3).

We find that stronger interaction-transmission association $\alpha$ leads to evolution of higher frequency of cooperation and increased population mean fitness. Nevertheless, when cooperation and defection coexist, $\alpha$ is expected to be reduced by natural selection, leading to extinction of cooperation and decreased population mean fitness (result 3.4; electronic supplementary material, figure S1). With $\alpha = 0$, the benefit of cooperation cannot facilitate its evolution; it can only succeed if horizontal transmission is biased in its favour.

Indeed, in our model, horizontal transmission plays a major role in the evolution of cooperation: increasing the transmission of cooperation, $T_A$, or decreasing the transmission of defection, $T_B$, facilitates the evolution of cooperation. However, the effect of oblique transmission is more complicated. When there is horizontal transmission bias in favour of cooperation, $T_A > T_B$, increasing the rate of oblique transmission, $1 - v$, will facilitate the evolution of cooperation. By contrast, when the bias is in favour of defection, $T_A < T_B$, higher rates of vertical transmission, $v$, are advantageous for cooperation, and the rate of vertical transmission must be high enough ($v > \hat{v}$) for cooperation to fix in the population.

Our deterministic model provides a good approximation to outcomes of simulations of a stochastic model with population structure in which individuals can only interact with and transmit to their neighbours. In these structured populations, interaction-transmission association arises owing to both social interactions and horizontal cultural transmission being local (figure 4).

Feldman *et al.* [16] studied the dynamics of an altruistic phenotype with vertical cultural transmission and a gene that modifies the transmission of the phenotype. Their results are very sensitive to this genetic modification: without it, the conditions for invasion of the altruistic phenotype reduce to Hamilton's rule. Further work is needed to incorporate such genetic modification of cultural transmission into our model. Woodcock [19] stressed the significance of non-vertical transmission for the evolution of cooperation and carried out simulations with Prisoner's Dilemma pay-offs but without horizontal transmission or interaction-transmission association ($\alpha = 0$). Nevertheless, his results demonstrated that it is possible to sustain altruistic behaviour via cultural transmission for a substantial length of time. He further hypothesized that horizontal transmission can play an important role in the evolution of cooperation, and our results provide strong evidence for this hypothesis.

To understand the role of horizontal transmission, we first review the role of *assortment*. Eshel & Cavalli-Sforza [10] showed that altruism can evolve when the tendency for *assortative meeting*, i.e. for individuals to interact with others of their own phenotype, is strong enough. Fletcher & Doebeli [11] further argued that a general explanation for the evolution of altruism is given by assortment: the correlation between individuals that carry an altruistic trait and the amount of altruistic behaviour in their interaction group (see also Bijma & Aanen [12]). They suggested that to explain the evolution of altruism, we should seek mechanisms that generate assortment, such as spatial structure, repeated interactions and individual recognition. Our results highlight another mechanism for generating assortment: an association between social interactions and horizontal transmission that creates a correlation between one's partner for interaction and the partner for transmission. This mechanism does not require repeated interactions, spatial structure, or individual recognition. We show that high levels of such interaction-transmission association greatly increase the potential for evolution of cooperation. With strong enough interaction-transmission association, cooperation can increase in frequency when initially rare even when there is horizontal transmission bias against it ($T_A < T_B$).

How does non-vertical transmission generate assortment? Lewin-Epstein *et al.* [20] and Lewin-Epstein & Hadany [21] suggested that microbes which induce their hosts to act altruistically can be favoured by selection, which may help to explain the evolution of cooperation. From the kin selection point-of-view, if microbes can be transmitted *horizontally* from one host to another during host interactions, then following horizontal transmission the recipient host will carry microbes that are closely related to those of the donor host, even when the two hosts are (genetically) unrelated. From the assortment point-of-view, infection by behaviour-determining microbes during interactions effectively generates assortment because a recipient of help may be infected by a behaviour-determining microbe and consequently become a helper. Horizontal cultural horizontal transmission can similarly generate assortment between cooperators and enhance the benefit of cooperation if cultural transmission and helping interactions occur between the same individuals, i.e. when there is interaction-transmission association, so that the recipient of help may also be the recipient of the cultural trait for cooperation. Thus, with horizontal transmission, 'assortment between focal cooperative players and cooperative acts in their interaction environment' [11, p. 15] is generated not because the helper is likely to be helped, but rather because the helped is likely to become a helper.

**Data accessibility.** This article has no additional data.

**Authors' contributions.** D.C.: conceptualization, formal analysis, investigation, methodology, software, visualization, writing-original draft, writing-review and editing; O.L.-E.: conceptualization, formal analysis, investigation, methodology, software, visualization, writing-original draft, writing-review and editing; M.W.F.: conceptualization, formal analysis, investigation, methodology, writing-review and editing; Y.R.: conceptualization, formal analysis, funding acquisition, investigation, methodology, project administration, supervision, visualization, writing-original draft, writing-review and editing

**Competing interests.** We declare we have no competing interests.

**Funding.** This work was supported in part by the Clore Foundation Scholars Programme (O.L.-E.), the Morrison Institute for Population and Resources Studies at Stanford University (M.W.F.), Israel Science Foundation (YR 552/19) and Minerva Stiftung Center for Lab Evolution (Y.R.)

**Acknowledgements.** We thank Lilach Hadany, Ayelet Shavit, Kaleda Krebs Denton and Tal Simon for discussions and comments.

## Endnotes

[1]In an extended model, which allows an individual to encounter $N$ individuals before choosing a partner, the right-hand side is multiplied by $E[N]$, the expected number of encounters ([10], eqn (4.6)).
[2]Inequality (1.3) generalizes inequalities (1.1) and (1.2) by substituting $p_C = r + p$, $p_D = p$ and $p_C = m + (1 - m)p$, $p_D = (1 - m)p$, respectively, where $p$ is the frequency of cooperators.

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
