## [Peer Review File · Proceedings of the Royal Society B: Biological Sciences]

Review History

RSPB-2020-3162.R0 (Original submission)

Review form: Reviewer 1

Recommendation

Accept with minor revision (please list in comments)

Scientific importance: Is the manuscript an original and important contribution to its field?

Excellent

General interest: Is the paper of sufficient general interest?

Good

Quality of the paper: Is the overall quality of the paper suitable?

Excellent

Is the length of the paper justified?

Yes

Should the paper be seen by a specialist statistical reviewer?

No

Do you have any concerns about statistical analyses in this paper? If so, please specify them explicitly in your report.

No

It is a condition of publication that authors make their supporting data, code and materials available - either as supplementary material or hosted in an external repository. Please rate, if applicable, the supporting data on the following criteria.

Is it accessible?

N/A

Is it clear?

N/A

Is it adequate?

N/A

Do you have any ethical concerns with this paper?

No

Comments to the Author

In “Non-Vertical Cultural Transmission, Assortment, and the Evolution of Cooperation,” the authors present a discrete-time dynamical system for the frequency of a cooperative trait in the presence of both social and genetic transmission. After defining the model and its parameters, the authors summarize its equilibria and their stability, presenting several conditions, in the form of inequalities, characterizing the various outcomes (fixation of one type or polymorphism). The social transmission parameter is then allowed to evolve via a modifier locus, and it is shown that the resulting dynamics favor lower population fitness.

Overall, I enjoyed reading the paper and I’m supportive of it being published in Proceedings B. One disadvantage of the model is that it has many parameters, but at the same time it seems to be about as simple as possible while capturing all of the qualitative features of interest. I also liked that the analysis done in the appendix is somewhat clean, and a relatively complete description of the evolutionary dynamics can be described in the main text. I do, however, have a number of specific comments, which can be found below.

Comments:

-References should be ordered by appearance if using bracketed numbers.

-The literature surrounding equations 1-4 is just a small portion of work related to conditions of this form. One obviously cannot cover everything, but studies of the effect of space on altruism often result in conditions of the form $b > \{\text{something}\} * c$. These include approaches using both inclusive fitness (e.g. “Evolution of cooperation in a finite homogeneous graph” by Taylor, Day, and Wild) as well as direct analyses of the spatial structure (e.g. “Evolutionary dynamics on any population structure” by Allen et al.). The text would benefit from some further discussion of these kinds of results (of the authors choosing – the above are just some examples).

-Around line 64, I find it odd to say that φ takes the role of relatedness. I understand that this is probably meant to compare the structural form of this inequality to Hamilton’s rule, but too often it seems that people try to suggest that something should be interpreted in terms of relatedness when it is not appropriate to do so. I would remove this sentence to prevent misunderstanding.

-At the end of line 82, the authors say that evolution of cooperation can be enhanced “partly because [cultural transmission] can diminish the effect of natural selection.” Later on in the paper,

the meaning of this claim becomes clearer, but it really left me scratching my head as to what it could mean when stated so briefly at the end of the introduction. It would be helpful to have a slightly more detailed explanation of what to anticipate from this statement.

-Around line 92, \tilde{p} looks nearly identical to \hat{p} unless one looks closely. Can the authors change these symbols so that they appear more distinct in the subsequent equations?

-Equation 8 needs to be described more clearly, especially if the paper is to be published in Proceedings B. A careful reading suggests that individuals interact in pairs and obtain payoffs (fitness) from these interactions. These payoffs are to be interpreted in a relative sense, with competition happening proportional to payoff (globally). There is then reproduction based on this competition, and the offspring inherits a trait based on horizontal transmission. Vertical transmission, which is described on the previous page, is then taken into account, which together with equation 8 gives a complete description of the dynamical system, equation 9. The latter (about vertical transmission) is briefly mentioned following equation 8, but I really think that the paper would benefit from several sentences—or even a paragraph—completely explaining all of the terms that appear in equation 8 and the underlying assumptions (e.g. infinite panmictic population versus sampling with stochasticity, etc.).

-In “Result 1” (line 130), “a cooperation” should be “cooperation.”

-Please discuss conditions 1-4 (lines 132-136) in more detail, as it relates to the mathematical properties of equilibria. (Might also be worthwhile to mention that “stable polymorphism” is sometimes called “coexistence” and “unstable polymorphism” is “bistable competition”).

-On line 144, by “unites” you just mean that the two conditions predict the same selection pressures locally, right?

-Line 160: “...this condition cannot be formulated in the form of Hamilton’s rule due to the bias in horizontal transmission...” This is an important point, as Hamilton’s rule applies to a rather special class of models. I wonder, though, what is the purpose of this statement here? I am not suggesting that the authors remove it, but rather that they explain why one might expect a version of Hamilton’s rule in this case.

-Later on, around line 166, the authors state that the quantity $\alpha T/(1-T)$ “can be regarded as the effective relatedness” in Hamilton’s rule. I don’t like this interpretation, as it seems to try to shoehorn something cultural into a biological statement. Sure, there are elements of both in the model, and this particular case pertains to the case $T_A = T_B$, but I still see very little value in describing this as an “effective relatedness,” and I think it could create confusion where there ought to be none.

-Line 192, please write “ $\text{sign}()$ ” in words in the main text, e.g. “the sign of [...] coincides with the sign of [...]”. It is fine to keep this notation in the appendix (as it is), though.

-Line 228: α must have been written as α since the symbol is not rendered.

-Line 242: The paragraph ending here is quite interesting, and I wonder whether a bit more intuition for the detrimental effects of the evolution of α can be provided here?

-In the section on population structure, the model setup had me a bit confused. Half of the population is chosen to initiate interactions, but are these really interactions or are they just actions? The game being considered is completely additive, which means a two-player interaction can be decomposed into two actions, one from each player. This confusion is compounded by the later expression for fitness as $w=1+b*n_b*c*n_c$, where both n_b and n_c are interaction counts. So if X and Y are neighbors and are both chosen to initiate an interaction, what happens if X chooses Y but Y chooses someone else? Does X still get a benefit from Y, or are these

expressions directional (as they seem like they should be)?

-Line 254: a “grid” seems like a lattice with a von Neumann neighborhood, but here the authors refer to a Moore neighborhood ($M=8$); is this correct? Maybe mention this up front as well.

-Line 274: “...under complex scenarios.” While it is true that the population is structured, in some ways it is nearly unstructured since it is so homogeneous. “Complex scenarios” to me would indicate something more topologically heterogeneous. I don’t think that the authors need to consider anything more complex here, but I would at least comment on the homogeneity of the primary structured population in the paper. A similar comment applies to lines 300-302 in the discussion section.

-In line 324, the authors claim that this mechanism does not require population structure. However, it feels like there is a version of population structure baked into the model via its parameters. I think I understand what is meant here, but some minor clarification would be nice.

-In the conclusion around line 336, what is being referred to is that the model effectively generates identity by state that does not come from identity by descent; is this correct?

-Figure 1: “showing The fitness” should be “showing the fitness”. Also, I believe that “prisoners’ dilemma” should be “prisoner’s dilemma” because the dilemma applies to the individual and not the group (though there is a conflict of interest between the two).

Review form: Reviewer 2

Recommendation

Accept with minor revision (please list in comments)

Scientific importance: Is the manuscript an original and important contribution to its field?

Good

General interest: Is the paper of sufficient general interest?

Good

Quality of the paper: Is the overall quality of the paper suitable?

Good

Is the length of the paper justified?

Yes

Should the paper be seen by a specialist statistical reviewer?

No

Do you have any concerns about statistical analyses in this paper? If so, please specify them explicitly in your report.

No

It is a condition of publication that authors make their supporting data, code and materials available - either as supplementary material or hosted in an external repository. Please rate, if applicable, the supporting data on the following criteria.

Is it accessible?

Yes

Is it clear?

Yes

Is it adequate?

Yes

Do you have any ethical concerns with this paper?

No

Comments to the Author

The paper studies the evolution of cooperation and interaction-transmission association using simulations and analytical methods. The study is carefully carried out and the manuscript is well-written.

I have one question about their choice of the payoff matrix. In fact, they use a simplified Prisoner's Dilemma -- so-called donation games, to model social interactions. How about using a general payoff matrix with R, S, T, P?

Also vertical transmission is not explicitly modeled using diploid sexual reproduction with recombination/mutation. How their results under simple assumption would change under more realistic vertical transmission?

Decision letter (RSPB-2020-3162.R0)

24-Mar-2021

Dear Dr Ram:

Your manuscript has now been peer reviewed and the reviews have been assessed by an Associate Editor. The reviewers' comments (not including confidential comments to the Editor) and the comments from the Associate Editor are included at the end of this email for your reference. As you will see, the reviewers and the Editors have raised some concerns with your manuscript and we would like to invite you to revise your manuscript to address them.

Research ethics:

Use of animals and field studies:

It is a condition of publication that you make available the data and research materials supporting the results in the article. Please see our Data Sharing Policies (<https://royalsociety.org/journals/authors/author-guidelines/#data>). Datasets should be deposited in an appropriate publicly available repository and details of the associated accession number, link or DOI to the datasets must be included in the Data Accessibility section of the article (<https://royalsociety.org/journals/ethics-policies/data-sharing-mining/>). Reference(s) to datasets should also be included in the reference list of the article with DOIs (where available).

Please submit a copy of your revised paper within three weeks. If we do not hear from you within this time your manuscript will be rejected. If you are unable to meet this deadline please let us know as soon as possible, as we may be able to grant a short extension.

Best wishes,
Dr Locke Rowe
mailto: proceedingsb@royalsociety.org

Editor

Comments to Author:

The board member notes that the terse style of the presentation is a bit of a departure from the more common style in the journal. While I am comfortable with the way it is now, the authors may wish to alter the presentation for greater impact on the typical audience of this journal.

Associate Editor

Board Member: 1

Comments to Author:

This manuscript presents a model for the evolution of cooperation when there is both vertical and nonvertical cultural transmission of the trait. The model also allows for different degrees of horizontal transmission between individuals who are undergoing fitness determining social interaction and, interestingly, it also allows this “interaction-transmission” association probability to evolve. I think the manuscript does an excellent job of providing some synthetic results for the general issue of the evolution of cooperation and the findings will likely be of interest to many of the readers of Proc B. Both referees were also strongly in favour of publication, although only one of these provided substantive comments. I agree with all of their suggestions and have two further comments of my own.

First, I think it is important to have a very clear description of the structure of the model and Figure 1 partly does this job. I wonder, though, if it can somehow incorporate the ordering of the different events in the life cycle more clearly as well. Related to this, L87 refers to Fig 1a when it should be Fig 1d.

Second, although I appreciate the rather terse, point-form sort of presentation in the manuscript, it does seem like a bit of a departure from the usual style of Proc B papers (which tend to have a more narrative structure). Indeed, it reads like a paper one might find in a more technical journal. This is an editorial issue though so I will leave it to the editor to decide if there is anything that should be done about this.

Reviewer(s)' Comments to Author:

Referee: 1

Comments to the Author(s)

In “Non-Vertical Cultural Transmission, Assortment, and the Evolution of Cooperation,” the authors present a discrete-time dynamical system for the frequency of a cooperative trait in the presence of both social and genetic transmission. After defining the model and its parameters, the authors summarize its equilibria and their stability, presenting several conditions, in the form of inequalities, characterizing the various outcomes (fixation of one type or polymorphism). The social transmission parameter is then allowed to evolve via a modifier locus, and it is shown that the resulting dynamics favor lower population fitness.

Overall, I enjoyed reading the paper and I'm supportive of it being published in Proceedings B. One disadvantage of the model is that it has many parameters, but at the same time it seems to be about as simple as possible while capturing all of the qualitative features of interest. I also liked that the analysis done in the appendix is somewhat clean, and a relatively complete description of

the evolutionary dynamics can be described in the main text. I do, however, have a number of specific comments, which can be found below.

Comments:

-References should be ordered by appearance if using bracketed numbers.

-The literature surrounding equations 1-4 is just a small portion of work related to conditions of this form. One obviously cannot cover everything, but studies of the effect of space on altruism often result in conditions of the form $b > \{\text{something}\} * c$. These include approaches using both inclusive fitness (e.g. "Evolution of cooperation in a finite homogeneous graph" by Taylor, Day, and Wild) as well as direct analyses of the spatial structure (e.g. "Evolutionary dynamics on any population structure" by Allen et al.). The text would benefit from some further discussion of these kinds of results (of the authors choosing – the above are just some examples).

-Around line 64, I find it odd to say that φ takes the role of relatedness. I understand that this is probably meant to compare the structural form of this inequality to Hamilton's rule, but too often it seems that people try to suggest that something should be interpreted in terms of relatedness when it is not appropriate to do so. I would remove this sentence to prevent misunderstanding.

-At the end of line 82, the authors say that evolution of cooperation can be enhanced "partly because [cultural transmission] can diminish the effect of natural selection." Later on in the paper, the meaning of this claim becomes clearer, but it really left me scratching my head as to what it could mean when stated so briefly at the end of the introduction. It would be helpful to have a slightly more detailed explanation of what to anticipate from this statement.

-Around line 92, \tilde{p} looks nearly identical to \hat{p} unless one looks closely. Can the authors change these symbols so that they appear more distinct in the subsequent equations?

-Equation 8 needs to be described more clearly, especially if the paper is to be published in Proceedings B. A careful reading suggests that individuals interact in pairs and obtain payoffs (fitness) from these interactions. These payoffs are to be interpreted in a relative sense, with competition happening proportional to payoff (globally). There is then reproduction based on this competition, and the offspring inherits a trait based on horizontal transmission. Vertical transmission, which is described on the previous page, is then taken into account, which together with equation 8 gives a complete description of the dynamical system, equation 9. The latter (about vertical transmission) is briefly mentioned following equation 8, but I really think that the paper would benefit from several sentences – or even a paragraph – completely explaining all of the terms that appear in equation 8 and the underlying assumptions (e.g. infinite panmictic population versus sampling with stochasticity, etc.).

-In "Result 1" (line 130), "a cooperation" should be "cooperation."

-Please discuss conditions 1-4 (lines 132-136) in more detail, as it relates to the mathematical properties of equilibria. (Might also be worthwhile to mention that "stable polymorphism" is sometimes called "coexistence" and "unstable polymorphism" is "bistable competition").

-On line 144, by "unites" you just mean that the two conditions predict the same selection pressures locally, right?

-Line 160: "...this condition cannot be formulated in the form of Hamilton's rule due to the bias in horizontal transmission..." This is an important point, as Hamilton's rule applies to a rather special class of models. I wonder, though, what is the purpose of this statement here? I am not suggesting that the authors remove it, but rather that they explain why one might expect a version of Hamilton's rule in this case.

-Later on, around line 166, the authors state that the quantity $\alpha T/(1-T)$ “can be regarded as the effective relatedness” in Hamilton’s rule. I don’t like this interpretation, as it seems to try to shoehorn something cultural into a biological statement. Sure, there are elements of both in the model, and this particular case pertains to the case $T_A = T_B$, but I still see very little value in describing this as an “effective relatedness,” and I think it could create confusion where there ought to be none.

-Line 192, please write “ $\text{sign}()$ ” in words in the main text, e.g. “the sign of [...] coincides with the sign of [...]”. It is fine to keep this notation in the appendix (as it is), though.

-Line 228: α must have been written as α since the symbol is not rendered.

-Line 242: The paragraph ending here is quite interesting, and I wonder whether a bit more intuition for the detrimental effects of the evolution of α can be provided here?

-In the section on population structure, the model setup had me a bit confused. Half of the population is chosen to initiate interactions, but are these really interactions or are they just actions? The game being considered is completely additive, which means a two-player interaction can be decomposed into two actions, one from each player. This confusion is compounded by the later expression for fitness as $w=1+b*n_b-c*n_c$, where both n_b and n_c are interaction counts. So if X and Y are neighbors and are both chosen to initiate an interaction, what happens if X chooses Y but Y chooses someone else? Does X still get a benefit from Y, or are these expressions directional (as they seem like they should be)?

-Line 254: a “grid” seems like a lattice with a von Neumann neighborhood, but here the authors refer to a Moore neighborhood ($M=8$); is this correct? Maybe mention this up front as well.

-Line 274: “...under complex scenarios.” While it is true that the population is structured, in some ways it is nearly unstructured since it is so homogeneous. “Complex scenarios” to me would indicate something more topologically heterogeneous. I don’t think that the authors need to consider anything more complex here, but I would at least comment on the homogeneity of the primary structured population in the paper. A similar comment applies to lines 300-302 in the discussion section.

-In line 324, the authors claim that this mechanism does not require population structure. However, it feels like there is a version of population structure baked into the model via its parameters. I think I understand what is meant here, but some minor clarification would be nice.

-In the conclusion around line 336, what is being referred to is that the model effectively generates identity by state that does not come from identity by descent; is this correct?

-Figure 1: “showing The fitness” should be “showing the fitness”. Also, I believe that “prisoners’ dilemma” should be “prisoner’s dilemma” because the dilemma applies to the individual and not the group (though there is a conflict of interest between the two).

Referee: 2

Comments to the Author(s)

The paper studies the evolution of cooperation and interaction-transmission association using simulations and analytical methods. The study is carefully carried out and the manuscript is well-written.

I have one question about their choice of the payoff matrix. In fact, they use a simplified Prisoner's Dilemma -- so-called donation games, to model social interactions. How about using a general payoff matrix with R, S, T, P?

Also vertical transmission is not explicitly modeled using diploid sexual reproduction with recombination/mutation. How their results under simple assumption would change under more realistic vertical transmission?

Author's Response to Decision Letter for (RSPB-2020-3162.R0)

See Appendix A.

Decision letter (RSPB-2020-3162.R1)

29-Apr-2021

Dear Dr Ram

I am pleased to inform you that your manuscript entitled "Non-Vertical Cultural Transmission, Assortment, and the Evolution of Cooperation" has been accepted for publication in Proceedings B.

Data Accessibility section

Open Access

Your article has been estimated as being 8 pages long. Our Production Office will be able to confirm the exact length at proof stage.

Paper charges

Sincerely,
Dr Locke Rowe
Editor, Proceedings B
mailto: proceedingsb@royalsociety.org

Appendix A

Cohen et al., RSPB-2020-3162

Reply to Editor

April 25, 2021

Dear Editor,

Below are the editors' and reviewers' comments in black and our answers in blue.

Editor

The board member notes that the terse style of the presentation is a bit of a departure from the more common style in the journal. While I am comfortable with the way it is now, the authors may wish to alter the presentation for greater impact on the typical audience of this journal.

- We thank the editor for giving us an opportunity to revise the manuscript. We feel that the current presentation, although being somewhat different from the usual style in the journal, is suitable for our results and conclusions. We did however make an additional effort to illustrate the model by modifying Figure 1, and to explain our results by adding more intuition to Result 1 (line 144).

Associate Editor

This manuscript presents a model for the evolution of cooperation when there is both vertical and nonvertical cultural transmission of the trait. The model also allows for different degrees of horizontal transmission between individuals who are undergoing fitness determining social interaction and, interestingly, it also allows this "interaction-transmission" association probability to evolve. I think the manuscript does an excellent job of providing some synthetic results for the general issue of the evolution of cooperation and the findings will likely be of interest to many of the readers of Proc B. Both referees were also strongly in favour of publication, although only one of these provided substantive comments. I agree with all of their suggestions and have two further comments of my own.

- We thank the editor for the feedback and comments.

First, I think it is important to have a very clear description of the structure of the model and Figure 1 partly does this job. I wonder, though, if it can somehow incorporate the ordering of the different events in the life cycle more clearly as well.

- We thank the reviewer for the suggestion. We modified Figure 1 to better explain the events of the life cycle. The figure now shows the two stages: Panel (a) describes transmission from parents or non-parental adults to juveniles and Panel (b) shows social interactions and horizontal transmission between adults. Panels (c) and (d) describe the payoff matrix and the horizontal transmission probabilities, respectively.

Related to this, L87 refers to Fig 1a when it should be Fig 1d.

- Fixed. We changed the order of panels in Fig 1, so the reference to Fig 1a is now correct (line 91)

Second, although I appreciate the rather terse, point-form sort of presentation in the manuscript, it does seem like a bit of a departure from the usual style of Proc B papers

(which tend to have a more narrative structure). Indeed, it reads like a paper one might find in a more technical journal. This is an editorial issue though so I will leave it to the editor to decide if there is anything that should be done about this.

- We thank the editor for giving us an opportunity to revise the manuscript. We feel that the current presentation, although being somewhat different from the usual style in the journal, is suitable for our results and conclusions. We did however make an additional effort to illustrate the model by modifying Figure 1, and to explain our results by adding more intuition to Result 1 (line 144).

Referee: 1

In “Non-Vertical Cultural Transmission, Assortment, and the Evolution of Cooperation,” the authors present a discrete-time dynamical system for the frequency of a cooperative trait in the presence of both social and genetic transmission. After defining the model and its parameters, the authors summarize its equilibria and their stability, presenting several conditions, in the form of inequalities, characterizing the various outcomes (fixation of one type or polymorphism). The social transmission parameter is then allowed to evolve via a modifier locus, and it is shown that the resulting dynamics favor lower population fitness.

Overall, I enjoyed reading the paper and I’m supportive of it being published in Proceedings B. One disadvantage of the model is that it has many parameters, but at the same time it seems to be about as simple as possible while capturing all of the qualitative features of interest. I also liked that the analysis done in the appendix is somewhat clean, and a relatively complete description of the evolutionary dynamics can be described in the main text.

- We appreciate the positive feedback and thank the reviewer for the comments, which we believe significantly improve the manuscript.

I do, however, have a number of specific comments, which can be found below.

Comments:

-References should be ordered by appearance if using bracketed numbers.

- Fixed

-The literature surrounding equations 1-4 is just a small portion of work related to conditions of this form. One obviously cannot cover everything, but studies of the effect of space on altruism often result in conditions of the form $b > \{something\} * c$. These include approaches using both inclusive fitness (e.g. “Evolution of cooperation in a finite homogeneous graph” by Taylor, Day, and Wild) as well as direct analyses of the spatial structure (e.g. “Evolutionary dynamics on any population structure” by Allen et al.). The text would benefit from some further discussion of these kinds of results (of the authors choosing—the above are just some examples).

- We added a new paragraph that introduces a result for evolution of cooperation on graphs (line 52 and new inequality 4):

“Cooperation can also evolve when interactions are determined by population structure. For example, Ohtsuki et al. [13] studied populations on graphs with average degree k , that is, the average individual has k potential interaction partners. Assuming that selection is weak and that the population size is much larger than k (i.e. sparse structure), they found that cooperative behaviour can evolve if

$$c < b \cdot 1/k . \quad (4)$$

They thus interpret $1/k$ as *social relatedness* or *social viscosity* [13].”

-Around line 64, I find it odd to say that φ takes the role of relatedness. I understand that this is probably meant to compare the structural form of this inequality to Hamilton’s rule, but too often it seems that people try to suggest that something should be interpreted in terms of relatedness when it is not appropriate to do so. I would remove this sentence to prevent misunderstanding.

- We changed line 65 from “ φ takes the role of relatedness (...) or assortment (...)” to “ φ replaces relatedness (...) or assortment (...)” to avoid incorrect interpretation.

-At the end of line 82, the authors say that evolution of cooperation can be enhanced “partly because [cultural transmission] can diminish the effect of natural selection.” Later on in the paper, the meaning of this claim becomes clearer, but it really left me scratching my head as to what it could mean when stated so briefly at the end of the introduction. It would be helpful to have a slightly more detailed explanation of what to anticipate from this statement.

- We added an explanation to our claim in order to make it clearer (line 85): “it diminishes the effect of selection (due to non-vertical transmission from non-reproducing individuals).”

-Around line 92, \tilde{p} looks nearly identical to \hat{p} unless one looks closely. Can the authors change these symbols so that they appear more distinct in the subsequent equations?

- To avoid confusion, we changed \tilde{p} to \dot{p} .

-Equation 8 needs to be described more clearly, especially if the paper is to be published in Proceedings B. A careful reading suggests that individuals interact in pairs and obtain payoffs (fitness) from these interactions. These payoffs are to be interpreted in a relative sense, with competition happening proportional to payoff (globally). There is then reproduction based on this competition, and the offspring inherits a trait based on horizontal transmission. Vertical transmission, which is described on the previous page, is then taken into account, which together with equation 8 gives a complete description of the dynamical system, equation 9. The latter (about vertical transmission) is briefly mentioned following equation 8, but I really think that the paper would benefit from several sentences—or even a paragraph—completely explaining all of the terms that appear in equation 8 and the underlying assumptions (e.g. infinite panmictic population versus sampling with stochasticity, etc.).

- We expanded the text following eq. 7, now eq. 8 in the revised manuscript (line 114):

“For example, the first term in Eq. 8 describes the case where two juveniles with phenotype A interact with probability \hat{p}^2 . In this case, the focal individual will retain its phenotype if (i) its social interaction partner is also its horizontal transmission partner, with probability α ; or (ii) its horizontal transmission partner is another individual, with probability $(1 - \alpha)$, and (ii.a) that individual also has phenotype A , with probability \hat{p} , or (ii.b) that individual has phenotype B , with probability $(1 - \hat{p})$, but horizontal transmission is unsuccessful, with probability $(1 - T_B)$. The frequency of A among parents follows a similar dynamic but must also include the effect of natural selection. Therefore, each right-hand term from Eq. 8 is multiplied by the corresponding fitness value (Table 1, Figure 1c), which depends on the phenotypes of the two interaction partners. Therefore, the frequency of phenotype A among parents is: ...”

- We changed the first line of the Models section (line 88) from “Consider a large population” to “Consider a very large well-mixed population” to emphasize the underlying assumptions.

-In “Result 1” (line 130), “a cooperation” should be “cooperation.”

- Fixed

-Please discuss conditions 1-4 (lines 132-136) in more detail, as it relates to the mathematical properties of equilibria.

- We added more details on conditions 1-4 in line 149:

“Thus, cooperation can take over the population if it has either a horizontal transmission advantage, or if it has a horizontal transmission disadvantage but the vertical transmission rate is high enough. In either case, the cost of cooperation must be small enough. A stable polymorphism can exist between cooperation and defection only if defection has a horizontal transmission advantage. In this case, the existence of a stable polymorphism depends on the interplay between the benefit and cost of cooperation and the vertical transmission rate.”

(Might also be worthwhile to mention that “stable polymorphism” is sometimes called “coexistence” and “unstable polymorphism” is “bistable competition”).

- We now mentioned this in line 155: “Note that stable and unstable polymorphism are also called, respectively, coexistence and bistable competition.”

-On line 144, by “unites” you just mean that the two conditions predict the same selection pressures locally, right?

- We are sorry for the confusion. By “unites” we meant that if we merge the conditions for fixation of cooperation and coexistence (stable polymorphism) we get ineq. 13 (formerly ineq. 12). We changed the text (line 163) to read “merges the conditions” instead of “unites the conditions”.

-Line 160: “...this condition cannot be formulated in the form of Hamilton’s rule due to the bias in horizontal transmission...” This is an important point, as Hamilton’s rule applies to a rather special class of models. I wonder, though, what is the purpose of this

statement here? I am not suggesting that the authors remove it, but rather that they explain why one might expect a version of Hamilton's rule in this case.

- To address the reviewer's comment, we changed line 179: we replaced "In general" with "importantly" and added "commonly used" to reflect that Hamilton's rule is commonly used in the literature and textbooks. This also corresponds to the manuscript's introduction. The text now reads (line 179):

"Importantly, this condition cannot be formulated in the commonly used form of Hamilton's rule due to the bias in horizontal transmission"

-Later on, around line 166, the authors state that the quantity $\alpha T/(1-T)$ "can be regarded as the effective relatedness" in Hamilton's rule. I don't like this interpretation, as it seems to try to shoehorn something cultural into a biological statement. Sure, there are elements of both in the model, and this particular case pertains to the case $T_A = T_B$, but I still see very little value in describing this as an "effective relatedness," and I think it could create confusion where there ought to be none.

- To avoid confusion, per the reviewer's comment, we replaced "effective" with "cultural". We consider "cultural relatedness" to reflect the probability that two individuals have the same cultural trait, much like genetic relatedness is the probability that two individuals have the same genetic trait. This is similar to the term "social relatedness" used by Ohtsuki et al. 2006.

The revised text now reads (line 184):

"This condition can be interpreted as a version of Hamilton's rule ($c < b \cdot r$, inequality 1) or as a version of inequality 3, where $\alpha T/(1-T)$ is a measure of *cultural relatedness* or *cultural assortment*, respectively, similar to the term *social relatedness* used by Ohtsuki et al. [13]."

- We also changed "effective relatedness" to "relatedness" in the abstract (line 20).

-Line 192, please write "sign()" in words in the main text, e.g. "the sign of [...] coincides with the sign of [...]." It is fine to keep this notation in the appendix (as it is), though.

- Fixed

-Line 228: α must have been written as α since the symbol is not rendered.

- Fixed

-Line 242: The paragraph ending here is quite interesting, and I wonder whether a bit more intuition for the detrimental effects of the evolution of α can be provided here?

- We have added two additional sentences (line 262) to give some intuition on why selection acts to reduce α :

"Note that α controls how often an individual learns from its interaction partner. However, from the *phenotype-centred view*, there is no incentive to do so: a cooperator interacting with a defector will not only pay the cost of cooperation but will also risk being "converted" to defection (with probability T_B), whereas a defector interacting with a cooperator will forfeit (with probability T_A) the benefit it received."

-In the section on population structure, the model setup had me a bit confused. Half of the population is chosen to initiate interactions, but are these really interactions or are they just actions? The game being considered is completely additive, which means a two-player interaction can be decomposed into two actions, one from each player. This confusion is compounded by the later expression for fitness as $w=1+b*n_b-c*n_c$, where both n_b and n_c are interaction counts.

- Indeed, half of the population is chosen to initiate interactions, each with a randomly chosen neighbour. Thus, the expected number of social interactions each individual participates in is 1, and the expected number of cultural transmission interactions is also 1. The number of interactions in which an individual participates may be zero (if it was not chosen to initiate interactions and was not selected by any of his neighbour initiators), one, or greater than one (if the individual was chosen to initiate an interaction and was also selected by a neighbour initiator for an interaction). We model interactions using this procedure to partition a population on a lattice into couples. This procedure was previously used by Lewin-Epstein et al. (Nat Comm 2017). Moreover, an individual that participates in more than one interaction may change its phenotype from one interaction to the other, due to horizontal transmission (see line 286). Therefore, we calculate for each individual the number of times it provided “help” and the number of times it received “help”.

We modified the text to make this clearer (line 272):

“In each generation, half of the individuals are randomly chosen to "initiate" interactions. These initiators interact (i) in a prisoner’s dilemma game with a random neighbour (i.e. individual in a neighbouring site); and (ii) in horizontal cultural transmission with a random neighbour (with replacement, i.e. possibly the same neighbour). The expected number of each of these interactions per individual per generation is one,, but the realized number of interactions can be zero, one, or even more than one, and in every interaction both individuals are affected, not just the initiator.”

So if X and Y are neighbors and are both chosen to initiate an interaction, what happens if X chooses Y but Y chooses someone else? Does X still get a benefit from Y, or are these expressions directional (as they seem like they should be)?

- Yes, X still gets a benefit from Y (if Y is a cooperator). Indeed, when X chooses Y, a symmetric interaction takes places between them (either social interaction, i.e. prisoner’s dilemma game, or cultural transmission). Later, Y might choose someone else to interact with, again in a symmetric interaction. That’s why an individual can take part in 0, 1, or even more than 1 interactions per generation, but with an expected value of 1. This is emphasized in line 276.

-Line 254: a “grid” seems like a lattice with a von Neumann neighborhood, but here the authors refer to a Moore neighborhood ($M=8$); is this correct? Maybe mention this up front as well.

- We indeed refer to a Moore neighbourhood. We now mention this in line 280: “On an infinite grid, $M=8$ (i.e. Moore neighbourhood [Moore 1962]),”

-Line 274: “...under complex scenarios.” While it is true that the population is structured, in some ways it is nearly unstructured since it is so homogeneous. “Complex scenarios”

to me would indicate something more topologically heterogeneous. I don't think that the authors need to consider anything more complex here, but I would at least comment on the homogeneity of the primary structured population in the paper. A similar comment applies to lines 300-302 in the discussion section.

- We revised the two places referred by the reviewer to avoid identifying the structured model as “complex”.

Line 298: “These comparisons show that the conditions derived for the deterministic unstructured model can be useful for predicting the dynamics in stochastic and structured models.”

Line 326: “Our deterministic model provides a good approximation to outcomes of simulations of a ~~complex~~ stochastic model with population structure...”

-In line 324, the authors claim that this mechanism does not require population structure. However, it feels like there is a version of population structure baked into the model via its parameters. I think I understand what is meant here, but some minor clarification would be nice.

- We agree that the alpha parameter introduces, in effect, population structure. To clarify this, we replaced “population structure” with “spatial structure” in line 350: “This mechanism does not require repeated interactions, spatial structure, or individual recognition.”

-In the conclusion around line 336, what is being referred to is that the model effectively generates identity by state that does not come from identity by descent; is this correct?

- This is correct, but because transmission in our model is cultural and possibly non-vertical, we do not use the term “by descent”.

-Figure 1: “showing The fitness” should be “showing the fitness”. Also, I believe that “prisoners’ dilemma” should be “prisoner’s dilemma” because the dilemma applies to the individual and not the group (though there is a conflict of interest between the two).

- Fixed.

Referee: 2

The paper studies the evolution of cooperation and interaction-transmission association using simulations and analytical methods. The study is carefully carried out and the manuscript is well-written.

- We thank the reviewer for the considerate evaluation.

I have one question about their choice of the payoff matrix. In fact, they use a simplified Prisoner's Dilemma -- so-called donation games, to model social interactions. How about using a general payoff matrix with R, S, T, P?

- That is correct. Line 101 now reads: “a cooperator suffers a fitness cost $\theta < c < I$, and its partner gains a fitness benefit b , where we assume $c < b$ (i.e. donation game).”

We focus on donation games because these games are commonly used in the literature on the evolution of cooperation and altruism (e.g. Hilbe et al. PNAS 2013). Specifically, we are interested in the case where cooperation is advantageous to the population but disadvantageous to the individual, which is the case in donation games, but not the case with a general payoff matrix.

Also vertical transmission is not explicitly modeled using diploid sexual reproduction with recombination/mutation. How their results under simple assumption would change under more realistic vertical transmission?

- In our paper we consider cultural transmission rather than genetic transmission, and, therefore, there is no concept of diploids or sexual reproduction. To emphasize this, we added “cultural” in line 89: “an offspring inherits its phenotype from its parent via cultural vertical transmission”. - -
- We assume cultural vertical transmission is uni-parental (line 92). In future work, we might assume bi-parental transmission (see Cavalli-Sforza & Feldman, 1981.)

Other changes:

- Removed comma after “i.e.” everywhere.
- Fixed typo in line 252 (“this is called...”)
- In line 252, enclosed “this is called external stability” in parentheses
- Italics instead of quotation marks: modifier allele (line 248) and external stability (line 252).
- “neighbor” replaced with “neighbour”; “neighborhood” replaced with “neighbourhood”